# Depletion of Lipocalin 2 (LCN2) in Mice Leads to Dysbiosis and Persistent Colonization with Segmented Filamentous Bacteria

**DOI:** 10.3390/ijms222313156

**Published:** 2021-12-05

**Authors:** Patrick Klüber, Steffen K. Meurer, Jessica Lambertz, Roman Schwarz, Silke Zechel-Gran, Till Braunschweig, Sabine Hurka, Eugen Domann, Ralf Weiskirchen

**Affiliations:** 1German Centre for Infection Research, Institute of Medical Microbiology, Justus-Liebig-University, D-35392 Giessen, Germany; patrick.klueber@ime.fraunhofer.de (P.K.); Silke.Zechel@mikrobio.med.uni-giessen.de (S.Z.-G.); 2Institute of Molecular Pathobiochemistry, Experimental Gene Therapy and Clinical Chemistry (IFMPEGKC), RWTH Aachen University Hospital, D-52074 Aachen, Germany; smeurer@ukaachen.de (S.K.M.); jessi.lambertz@freenet.de (J.L.); 3Labor Mönchengladbach, Medical Care Centre, D-41169 Mönchengladbach, Germany; RSchwarz@labor-stein.de; 4Institute of Pathology, RWTH Aachen University Hospital, D-52074 Aachen, Germany; tbraunschweig@ukaachen.de; 5Institute for Insect Biotechnology, Justus-Liebig-University, D-35392 Giessen, Germany; sabine.hurka@innere.med.uni-giessen.de; 6German Centre for Infection Research, Institute of Hygiene and Environmental Medicine, Justus-Liebig-University, D-35392 Giessen, Germany

**Keywords:** adipokine, lipocalin, NGAL, inflammation, gut, small intestine, microbiota, siderophore, segmented filamentous bacteria (SFB), mast cell

## Abstract

Lipocalin 2 (LCN2) mediates key roles in innate immune responses. It has affinity for many lipophilic ligands and binds various siderophores, thereby limiting bacterial growth by iron sequestration. Furthermore, LCN2 protects against obesity and metabolic syndrome by interfering with the composition of gut microbiota. Consequently, complete or hepatocyte-specific ablation of the *Lcn2* gene is associated with higher susceptibility to bacterial infections. In the present study, we comparatively profiled microbiota in fecal samples of wild type and *Lcn2* null mice and show, in contrast to previous reports, that the quantity of DNA in feces of *Lcn2* null mice is significantly lower than that in wild type mice (*p* < 0.001). By using the hypervariable V4 region of the 16S rDNA gene and Next-Generation Sequencing methods, we found a statistically significant change in 16 taxonomic units in *Lcn2*^-/-^ mice, including eight gender-specific deviations. In particular, members of *Clostridium*, *Escherichia*, *Helicobacter*, *Lactococcus*, *Prevotellaceae_UCG-001* and *Staphylococcus* appeared to expand in the intestinal tract of knockout mice. Interestingly, the proportion of *Escherichia* (200-fold) and *Staphylococcus* (10-fold) as well as the abundance of intestinal bacteria encoding the LCN2-sensitive siderphore enterobactin (*entA*) was significantly increased in male *Lcn2* null mice (743-fold, *p* < 0.001). This was accompanied by significant higher immune cell infiltration in the ileum as demonstrated by increased immunoreactivity against the pan-leukocyte protein CD45, the lymphocyte transcription factor MUM-1/IRF4, and the macrophage antigen CD68/Macrosialin. In addition, we found a higher expression of mucosal mast cell proteases indicating a higher number of those innate immune cells. Finally, the ileum of *Lcn2* null mice displayed a high abundance of segmented filamentous bacteria, which are intimately associated with the mucosal cell layer, provoking epithelial antimicrobial responses and affecting T-helper cell polarization.

## 1. Introduction

Lipocalin 2 (LCN2) is a small, 24-kDa secreted extracellular protein, which was originally purified from phorbol myristate acetate-stimulated human neutrophils [1]. Therefore, this protein is often referred to as neutrophil-gelatinase-associated lipocalin (NGAL). Structurally, it belongs to the lipocalin protein family sharing a highly compact hydrophobic β-barrel fold adapted to bind and transport a variety of hydrophobic ligands. Altered LCN2 expression occurs under diverse pathological conditions, including kidney disease, obesity, and acute and chronic inflammatory liver disease [2,3,4]. Upon intoxication, infection, inflammation, and other forms of cellular stress, LCN2 is rapidly induced in injured hepatocytes [5]. As a characteristic of an acute phase protein, LCN2 expression is stimulated by inflammatory cytokines including IL-6, IL-1β, and TNF-α [6,7]. In the intestine, inflammation leads to a profound higher expression of LCN2, which occurs, for example, in the chronic diseases colitis ulcerosa and Crohn’s disease (CD) [8,9]. Bacterial infection and upregulation of LCN2 are mutually linked since application of lipopolysaccharides boosts LCN2 expression [10].

Intoxication by Indomethacin in mice causes a reduction in intestinal length and body weight, LCN2 upregulation, elevated microscopic and macroscopic damage, and a significant reduction in expression of α-defensin and lysozyme in Paneth cells. In addition, there is a global change in the relative abundance of microbial communities [8]. These are classical signs of ileitis typically found in CD [11]. In particular, there is a characteristic change in the ratio of the phyla *Firmicutes* and *Bacterioidetes* as well as an increase in *Enterobacteriaceae* associated with the pathogenesis of CD [12,13]. The observed dysbiosis usually associated with CD contributes to the initiation and perpetuation of chronic mucosal inflammation [14,15,16].

In order to prevent inflammation and keep up the coexistence of the intestinal microbiota in the inflammatory free host mucosal compartment, there is a physiological barrier. This intestinal barrier comprises the mucus and buildup of mucins secreted by goblet cells [17]. Moreover, antimicrobial peptides, e.g., α-defensins and lysozyme, are secreted products of the Paneth cells in the crypts of the small intestine (distal ileum) [18,19]. Antimicrobial peptides are innate immunomodulators, which selectively control inflammation by the regulation of the microbial status of the intestine [20]. In addition to microbial imbalance, dysfunction of Paneth cells is also a characteristic feature of CD [21]. In CD patients, reduced expression of α-defensins results in reduced barrier function and impaired luminal antibacterial host defense capacity [22,23,24,25]. As mentioned above, bacterial infection leads to enhanced expression of LCN2 to protect the intestinal barrier function [6]. In turn, the loss of LCN2 has been shown to cause dysbiosis, provoking intestinal inflammation [26]. Despite higher expression in goblet and Paneth cells in active CD mutually influencing their barrier function, LCN2 itself directly participates in the host defense response [3,9].

Furthermore, LCN2 plays an eminent role in cellular iron metabolism. It has the ability to sequester iron-binding bacterial siderophores, thereby preventing bacterial access to iron and limiting their growth [27]. Detailed structural studies have demonstrated that LCN2 can bind to a broad range of catecholate siderophores chemically similar to enterobactin (Ent) and soluble siderophores of mycobacteria (i.e., carboxymycobactins), which are chemically distinct from Ent [28]. In line, the complete or hepatocyte-specific ablation of *Lcn2* in mice resulted in increased sensitivity to bacterial infections [26,29]. Conversely, the administration of LCN2 significantly inhibited the outgrowth of enteric *Escherichia coli* under both in vitro and in vivo conditions, suggesting a function in shaping microbiota homeostasis [10]. Similarly, *Lcn2* was shown to protect from colonic inflammation and intestinal carcinogenesis by controlling the bacterial community composition of the intestinal microbiota in the IL-10 knockout model of colitis [30]. In particular, mice lacking functional *Lcn2* exhibited elevated levels of gut bacteria expressing EntA compared to wild type littermates [6]. Under physiological conditions, the scavenging of Ent by LCN2 inhibits the growth of pathogenic bacteria that rely on Ent to scavenge iron from host cells [31]. Therefore, LCN2 was assumed to be a bacterially-induced protein playing important roles in maintaining gut homeostasis, thereby reducing the susceptibility to develop colitis [6].

We comparatively analyzed the diversity and genus spectrum of the intestinal microbiome of *Lcn2*^-/-^ and wild type mice. The relative abundance of conspicuously altered genera between both genotypes was determined by Next-Generation Sequencing (NGS) methods using the hypervariable V4 region of the 16S rDNA gene as a discriminator. Immunohistochemical staining of the small intestine revealed that the lack of *Lcn2* is associated with increased intestinal inflammatory activity, using the pan-leukocyte marker CD45, the multiple myeloma oncogene 1 (MUM-1) specific for B-cells, and the macrophage marker CD68. In addition, reverse transcription quantitative PCR (RT-qPCR) showed a higher frequency of mucosal mast cells (mMCs) in the terminal ileum of *Lcn2*-deficient animals. Most strikingly, we found a much higher abundance of segmented filamentous bacteria (SFB) in the content of the distal ileum and associated with the ileal epithelium. The SFB colonization was accompanied by an increase in epithelial-derived inducible NO synthase (NOS2), serum amyloid A1 and A2 (SAA1/2), and the antibacterial lectin regeneration islet-derived protein 3 γ (RegIIIγ), which was accompanied by an induction of T helper 17 (Th17) cell differentiation.

## 2. Results

### 2.1. Gender-Specific Alteration in the Quantity and Quality of Intestinal Microbiota in Lcn2 Null Mice

Previous studies indicate that the disruption of the *Lcn2* gene provokes changes in the intestinal microbial community. In particular, some potentially pathogenic bacterial species were identified to contribute to the shift in the species spectrum [6,32]. However, how growth and density of the bacterial population behave in this context is largely unknown. To comparatively study the intestinal bacterial load in wild type and *Lcn2* null mice, we first extracted total DNA (pro- and eukaryotic) from murine feces of both genotypes (Figure 1). Quantification of the obtained DNA showed that feces of *Lcn2* null mice contained significantly lower total DNA quantities than wild type animals (Figure 1A).

To further characterize the bacterial colonization of the murine intestinal tract, the extracted total DNA was next analyzed by quantitative PCR (qPCR) using *entA*-specific oligonucleotides as primers encoding an enzyme involved in the biosynthesis of the *Enterobacteriaceae*-typical siderophore enterobactin, which can be sequestered by LCN2 (Figure 1B). In this analysis, the efficiency of the *entA* primers (E = 105.3%) was determined by means of the slope parameter (m = −3.2) of an *Escherichia coli* standard curve.

We found that in the feces of male wild type mice, *entA* is seven times more abundant than in females (*p* < 0.05), whereas the gene in *Lcn2*-deficient animals is more than 4800 times higher abundant in males than in females (*p* < 0.001), indicating (i) a clear gender-specific difference and (ii) a strong increase in bacterial abundance if the iron supply is not limited by scavenging the siderophore, underlining the importance of LCN2.

The gender-specific quantification in female wild type and knockout mice did not show a statistically significant difference, showing only a low *entA* copy count (0.025 and 0.027 pg/µL). In contrast, the abundance of the gene in *Lcn2*^-/-^ males increased significantly (~743-fold; *p* < 0.001) compared to wild type mice of the same sex. These results imply a higher basal abundance of *entA* in males and a more pronounced increase in the absence of LCN2.

To next characterize the bacterial and fungal spectrum in fresh feces that can be cultivated aerobically or microaerobically, we dispersed the respective probes in physiological saline solution and plated them onto Columbia, MHF, MacConkey No. 3 and Sabouraud glucose media (Figure 2).

Phenotypically different bacterial colonies were observed in each sample, which were then picked and cultured in pure culture medium to identify them by mass spectrometry or 16S rDNA sequencing to the species level (Table 1). Despite several weeks of incubation of the Sabouraud glucose nutrient medium, no fungal growth was detected.

The combination of both methods made it possible to identify seven bacterial species with a confidence value or sequence identity of ≥98%. These are representatives of the genera *Escherichia*, *Staphylococcus*, *Aerococcus*, *Enterococcus*, *Lactococcus, Lactobacillus*, and *Streptococcus*. An eighth pure culture with 99% sequence identity could be assigned to the genus *Lactobacillus*, but the species level could not be determined unambiguously due to the high similarity of *Lactobacillus murinus* and *Lactobacillus animalis*.

The species *Escherichia coli, Staphylococcus xylosus*, and *Aerococcus viridans* were equally represented as core components in all four groups, whereas *Enterococcus faecalis* and *Lactobacillus murinus*/*Lactobacillus animalis* (wild type male), *Lactobacillus johnsonii* (knockout female), as well as *Lactococcus formosensis* and *Streptococcus danieliae* (knockout male) were detected exclusively in specific groups. In our analysis, the limitation of the MALDI-TOF mass spectrometry reference database (VITEK^®^ MS platform), which is mainly based on clinical isolates of human origin, could be compensated by the identification using 16S rDNA sequencing.

After the bioinformatic processing of the raw NGS data, a metric multidimensional scaling, i.e., principal coordinates analysis (PCoA), was carried out to deduce the extent of inter-individual or group-related differences. This analysis revealed that in contrast to the wild type samples, which closely clustered both at the group level and within one sex, there was a much greater distance noted between the samples of male and female knockout mice (Figure 3). Moreover, the slightly scattered position of the five male samples was particularly striking, indicating greater variation within this group, whereas the data points of the females clustered closely together, similar to those of the wild type mice.

### 2.2. The Observed Intestinal Microbiota Diversity between Wild Type and Lcn2 Null Mice Is Homogenous

Diversity is an important descriptive parameter providing information on the number of taxonomic units within an ecological community. The α diversity according to Whittaker describing the genus diversity within a habitat showed a total arithmetic mean of about 46 genera per sample, from which all groups differed only slightly (Figure 4A). In samples from male wild type mice, an average of 44 genera were detected and in those of the females about 47. In the *Lcn2*^-/-^ strain, similar to the wild type, the intestinal tract of female animals had a slightly higher diversity (n = 48) compared to males (n = 45), whereby both the genotypic and the sex-specific differences were weak and not statistically significant (*χ*^2^ = 3.26; *p* > 0.1).

After showing that the diversity of the groups was not significant different from each other, we next examined the genus spectrum in dependence of genotype and sex. The obtained results are depicted in Venn diagrams, showing that of a total of 68 different genera, 42 (~62%) were prevalent in all four groups, independent of sex and genotype. In addition, another 12 genera (~18%) could be detected in at least three groups. In contrast, 14 genera (20%) were only identified in two or exclusively in a single group (Figure 4B).

When comparing the gender-specific differences within the same genotype groups, striking differences were identified that were already noticed in the PCoA. While both wild type groups showed high similarity (~88%) in the genus spectrum (*I_J_* = 0.86), male and female *Lcn2*^-/-^ mice had a total similarity of 81.5% (*I_J_* = 0.8). In contrast, the genus spectrum of male mice of both genotypes differed strongly (*I_J_* = 0.32). Female mice of both genotypes only had a Jaccard index of *I_J_* = 0.19. However, it is particularly noticeable that knockout mice with 65 genera were populated with most of the microbial community present within the total spectrum (~96%), whereas only 57 genera (~84%) could be identified in the wild type animals.

### 2.3. Distribution and Abundance of Identified Genera

To further characterize the intestinal microbial community of *Lcn2* null and wild type mice, the relative frequency of all identified genera was determined. The data of the 16S rRNA amplicon sequencing comprised 4,045,823 reads, of which about 200,000 (~5%) could not be classified taxonomically at least to the family level. The summary of these sequencing results are depicted graphically in Figure 5. For this purpose, the relative abundances of each sample were summed groupwise to better illustrate group-related patterns and differences. In addition, the biological multiple approaches determined by next-generation sequencing are presented in Appendix A. A total of 68 different genera could be identified from the feces with a fixed detection limit of three reads per group. Rarefaction curves suggested a sufficient sequencing depth (Appendix A).

In our analysis, the *Bacteroidales_S24-7_group*, *Lachnospiraceae*, and *Lactobacillus* alone accounted for between 55% (female) and 68% (male) of total reads in feces of wild type mice, whereas in both *Lcn2*^-/-^ groups, four genera each (≥ 11%) achieved a cumulative proportion of more than 50%. At 11.3%, *Prevotella* complemented the three already mentioned genera in female knockout mice. More than three quarters of the total reads in male *Lcn2* null mice could be assigned to representatives of the *Bacteroidales_S24-7_group*, *Escherichia*, *Lactobacillus* and *Staphylococcus*. The expansion of *Staphylococcus* in the intestine of male mice was particularly striking, at about 41.4%. In contrast, representatives of the *Bacteroidales_S24-7_group* represented the most dominant group of bacteria both in the wild type (female 25.6%, male 36.7%) and in the intestine of female knockout mice (18.7%). In our study, the distribution of *Alistipes* was contrary to the findings of a previous study [6], possibly reflecting its expansion in the context of inflammatory intestinal processes in *Lcn2* null mice.

Genera with genotype-related and conspicuously altered frequency distribution were grouped into two groups, which are highlighted in color in Figure 5. *Alloprevotella*, the *Bacteroidales_S24-7_group*, *Coprococcus*, *Lachnospiraceae*, *Mollicutes_RF9*, *Rhodospirillum*, the *Ruminococcaceae_NK4A214_group*, *Ruminococcaceae_UCG*, *Ruminococcus*, and *Turicibacter* that were decreased or completely absent in the intestinal tract of *Lcn2*^-/-^ mice are highlighted in green. Genera missing in the wild type or that were conspicuously increased in *Lcn2*^-/-^ are highlighted in red.

These include *Clostridium*, *Escherichia*, *Helicobacter*, *Lactococcus*, *Prevotellaceae_UCG-001*, and *Staphylococcus*. In a next step, we tested individual differences for their statistical significance (Figure 6).

Representatives of the genera *Escherichia* (11.5%) and *Staphylococcus* (41.4%) were more significantly (*p* < 0.05) represented in male knockout mice than in wild type and female knockouts. The increase in relative abundance of *Prevotellaceae_UCG-001* (female: 1.1%, male: 0.42%) and *Lactococcus* (female: 0.63%, male: 1.9%) in *Lcn2*^-/-^ mice was also statistically significant (*p* < 0.05). The same applied to the obtained gender differences within the *Lcn2* null mice, whereas the proportion of *Prevotellaceae_UCG-001* was higher by a factor of 2.6 in females than in males. In contrast, the proportion of the genera *Escherichia* (~200-fold), *Lactococcus* (~3-fold), and *Staphylococcus* (~10-fold) in male mice was higher than that in females. *Clostridium* (female: 0.39%, male: 0.15%) and *Helicobacter* (female: 0.22%, male: 0.12%) were more abundant in the intestine of knockout mice compared to wild type mice (*p* < 0.05). However, no statistically significant gender differences were found.

The relative abundance of *Alloprevotella*, the *Bacteroidales_S24-7_group, Coprococcus, Lachnospira, Mollicutes_RF9, Rhodospirillum*, the *Ruminococcaceae_NK4A214_group, Ruminococcaceae_UCG*, and *Ruminococcus* was significantly decreased in the *Lcn2* nulls compared to the wild type (*p* < 0.05), whereas the degree of group-related significances varied depending on gender. In male *Lcn2*^-/-^ mice, the genera *Rhodospirillum, Ruminococcus*, and *Turicibacter* were completely absent. Despite large differences in *Turicibacter* levels between male and female wild type mice, no genotype specific significances were found. In *Lcn2* nulls, the decrease in female (~65-fold) mice was statistically proven (*p* < 0.05). The gender-specific differences in *Coprococcus*, the *Ruminococcaceae_NK4A214_group*, and *Turicibacter* within the wild type mice were significant (*p* < 0.05). *Turicibacter* was the only genus that was more abundant in female wild type mice than in males. The decrease in the relative abundance of *Coprococcus, Lachnospira, Rhodospirillum*, and *Ruminococcaceae_UCG* in the mutant had also gender-specific characteristics. The 13:1 ratio of the genus *Lachnospira* (~11.6%), which was strongly represented in female knockout mice, was particularly striking. This genus colonized the male intestine much more weakly (~0.9%) and its proportion was decreased 26 times compared to the wild type (23.4%). The *Bacteroidales_S24-7_group* and *Ruminococcus* showed no statistically significant gender-specific differences. In addition, the proportion of the *Bacteroidales_S24-7_group* in the knockout with 13.3% (male) and 18.7% (female) was still higher than 10% of the total reads and decreased by a factor of 2.8 and 1.4, respectively.

In a box plot using the genus *Escherichia* as an example, these fundamental differences in microbial composition are particularly exemplified. In this diagram, the values of the wild type lies together in a narrow range and have a small scattering width, whereas the interquartile distance of both knockout groups is significantly larger, with 0.029% (female) and 4.237% (male) (Figure 7).

### 2.4. LCN2 Deficiency Leads to an Increase in Cecal Weight

We next asked if the striking microbial alterations we observed had any impact on the physiology of the gastrointestinal tract. In order to investigate this, we first isolated the small and large intestinal tracts of wild type and *Lcn2* null mice (Figure 8). This analysis showed that the body and cecal weight of the *Lcn2* null mice were significantly increased compared to wild type mice (Figure 8A). Moreover, the body/cecal ratio was decreased in *Lcn2* null mice compared to wild type mice, implying that the cecal weight was disproportionally elevated. The length of the small intestine was slightly increased whereas the length comparison of the large intestine (cecum and colon) of the *Lcn2* null mice and wild type mice showed no difference (Figure 8B). To demonstrate *Lcn2* expression in the wild type and the absence of *Lcn2* mRNA and protein in knockout animals, we isolated RNA and proteins from the distal part of the small intestine (distal ileum), cecum, and the distal part of the large intestine. RT-qPCR (Figure 8C, *upper part*) and Western blot analysis (Figure 8C, *lower part*) confirmed expression of LCN2 in all three compartments of WT animals, which was lost upon knockout. This downregulation had no impact on the expression of the epithelial-to-mesenchymal transition (EMT) marker E-cadherin (Figure 8C, *lower part*).

### 2.5. LCN2 Deficiency Triggers the Adaptive and Innate Immune Response in the Small Intestine

In order to investigate if the dysbiosis of the digestive tract is associated with histological alterations, we next prepared paraffin sections from the small intestine (Figure 9A,B) stained with hematoxylin & eosine (H & E), Periodic acid-Schiff (PAS), chloroacetate esterase (CAE), and Giemsa, showing that the intestinal villi shape in the *Lcn2* null mice was somewhat less slender and tall. In addition, the CAE stain indicated more neutrophil polymorphs within the respective tissues sections of the *Lcn2* nulls. However, there was no gross macroscopic damage visible.

The staining in Figure 9A showed a higher cellularity in the villi of the *Lcn2* null animals compared to the wild type counterparts. To further visualize if there was higher inflammatory activity in the small intestine, we comparatively immunostained tissue sections of both genotypes with antibodies specific for the leukocyte common antigen (CD45), the lymphocyte-specific transcription factor MUM-1 also known as multiple myeloma oncogene 1 or interferon regulatory factor 4 (IRF4), and monocyte lineage marker CD68 (Figure 9B). All three markers were significantly higher expressed in the villi of the small intestinal tract of *Lcn2* null mice, indicating that the inflammatory activity was indeed significantly increased in the respective mice (Figure 9B). In a next step, we differentiated between the different CD4^+^ T-derived cells including Th1, Th2, T_reg_, and Th17 cells (Figure 9C, and below). The mRNA of the Th1 cell key transcription factor *T-bet*, in line with the cytokines *Tnf-α* and *Ifn-γ*, were elevated. In contrast, the mRNA encoding the transcription factor *Gata3* showed a tendency to be reduced, whereas the Th2 corresponding cytokines *Il-5* and *Il-13* were significantly downregulated in *Lcn2*-deficient animals in comparison to controls (Figure 9C). The expression of the T_reg_ relevant cytokine *Tgf-β1* as well as the key transcription factor *Foxp3* was not changed (Figure 9C). Due to the substantial changes in the bacterial colonization, mast cells as a professional defense mechanism of the innate immune response against bacteria were analyzed (Figure 9D–F). We included ear tissue of *Lcn2* null mice in our analysis, which was tested to be negative for *Lcn2* expression (Figure 9D, *upper*) and showed a high abundance of connective tissue mast cells (ctMC) when stained with Toluidine Blue (Figure 9F, *left*). The expression of tryptase (*Mcpt6*), indicative of ctMC, was markedly higher in ears compared to tissue of the distal ileum as evidenced in RT-qPCR, and the expression was not different between *Lcn2* null mice and respective controls (Figure 9D, *lower*). Moreover, Toluidine Blue stain of the distal ileum revealed that MC were only present in the connective tissue but not in the mucosal layers (Figure 9F, *right*). However, the expression of chymases (*Mcpt1*, *Mcpt2*), indicative of mucosal MC (mMC) and expressed at low levels in ear tissue (Figure 9E, *left*), was significantly higher in the distal ileum, especially in the distal ileum of *Lcn2* null mice compared to wild type animals (Figure 9E).

### 2.6. Loss of LCN2 Leads to Elevated and Persistant Colonization with Segmented Filamentous Bacteria

Finally, having a closer look at the content and the epithelium proximal structures in Giemsa and H & E stain, we noticed rod-like segmented structures in the ileum of *Lcn2* null mice that were not/or only very sparsely found in age-matched wild type mice (Figure 10A). These were associated with signs of autolysis in apical villi and detachment of epithelial cells from the basal membrane. In addition, those rod-like segmented bacteria showed a very intimate association with the epithelial cells, reaching through the mucus directly contacting the cells (Figure 10A, *red arrows*). Inspection of the extracted content of the distal ileum revealed the presence of those bacteria in the stool of *Lcn2* null mice (~70 days old, Figure 10B, *white arrows*). Whereas these structures were observed in all *Lcn2* null mice, those bacteria were only sparsely seen in ~50-day-old wild type mice but not at all in ~70-day-old wild type mice. Closer examination of these bacteria indicated that they were bipolar, having a bold, nearly unsegmented end (Figure 10C, *red circles*) and a fine, highly segmented tip (Figure 10C, *yellow circles*). A Gram stain of the distal ileal content of *Lcn2* null and wild type mice underpinned (i) the high frequency of those rod-shaped bacteria in *Lcn2* null mice and (ii) demonstrated that these rod-shaped bacteria were Gram-positive (Figure 10D, *white arrows*).

Since the above-mentioned criteria resembled those of SFB, we performed PCR with DNA extracted from the distant ileal content (Figure 11A). Those PCRs (i) validated the extraction of bacterial DNA and (ii) showed the presence and different degrees of colonization of the distant ileum of *Lcn2* null mice with SFB compared to wild type mice (Figure 11A). In this analysis, specific detection of SFB 16S rDNA was performed using three primers in two different combinations [33]. The identity of the resulting DNA amplicons specific for the V3 16S rRNA and the SFB-specific 16S rRNAs was confirmed by sequencing (not shown).

As a consequence of SFB colonization, their presence and intimate relationship with epithelial cells provoked several responses in those cells. For example, we could show by RT-qPCR that *iNos* (*Nos2*), *RegIIIγ*, a bactericidal C-type lectin that specifically targets Gram-positive bacteria, as well as the serum amyloid A proteins 1 (SAA1) and 2 (SAA2) were highly expressed in *Lcn2* null mice compared to wild type mice (Figure 11B). Downstream, these responses of epithelial cells impact the polarization of T-cells, leading to an expansion of Th17 helper cells. To this end, we could show by RT-qPCR that the Th17 related transcription factor receptor-related orphan receptor γt (*Rorγt*) as well as the Th17 cytokine profile including *Il-17a* and *Il-21* were highly expressed in *Lcn2* null mice compared to wild type mice, while the *IL-22* mRNA expression showed only a tendency to be upregulated (Figure 11C).

## 3. Discussion

In the present study, we show that the loss of *Lcn2* is associated with significant changes in the intestinal microbiome composition and increased inflammatory activity in the gastrointestinal tract. This confirms previous studies showing that *Lcn2* is a crucial component of the innate immune system mediating protective immune responses [3,34]. As such, elevated expression of LCN2 has been shown to be protective against bacterial infection because of its high affinity for siderophores. These are small, high-affinity iron-chelating compounds synthesized and secreted by microorganisms such as bacteria and fungi, serving primarily to transport iron across their cell membranes. Such strategies limiting nutrient accessibility of commensal and pathogenic microorganisms to essential elements are collectively termed as nutritional immunity [34]. LCN2 has a trilobite structure with pockets suitable for selective binding and sequestering of tris *ortho*-catechols, such as enterobactin/enterochelin, hydroxyphenyl oxazoline, heterocycle hydroxamate, and linear hydroxamate groups used by enteric bacteria [35].

As an acute phase protein, LCN2 is significantly overexpressed during the inflammatory response, thereby preventing bacterial growth and sequestering iron, which might cause unwanted oxidative stress and injury [34,35]. Consequently, mice lacking *Lcn2* are highly susceptible to bacterial infections [26,29].

We now show that under physiological conditions, respective mice without any additional challenge have dysbalances in intestinal homeostasis (Figure 12). This in turn provokes an expansion of siderophore-dependent species, leading to a shift in the genus spectrum and abundance, which favors fungal growth, modified metabolite synthesis, and failure in endothelial barrier function, leading to systemic bacterial translocation triggering hepatic and intestinal inflammation.

This concept is strongly supported by our findings. In our study, we showed that untreated *Lcn2*^-/-^ mice were extremely prone to bacterial infections. In addition, our microbiome analysis showed that the relative abundance of several bacterial genera including *Alloprevotella*, the *Bacteroidales_S24-7_group*, *Coprococcus*, *Lachnospira*, *Mollicutes_RF9*, *Rhodospirillum*, the *Ruminococcaceae_NK4A214_group*, *Ruminococcaceae_UCG*, *Ruminococcus*, and *Turicibacter* were significantly decreased in mice lacking LCN2. Several of these alterations were gender-dependent. Dysbiosis was further associated with increased inflammatory activity in the intestine as demonstrated by increased immunoreactivity against CD45, MUM-1, and CD68.

In previous studies, we found that *Lcn2* null mice showed significantly more liver damage after challenge with hepatotoxic substances or surgical procedures inducing hepatic damage [5]. Other groups have shown in models of inflammatory bowel disease that LCN2 is a host defense protein protecting against intestinal inflammation by controlling the bacterial community [30]. In particular, the facultative pathogenic *Alistipes* utilizing enterobactin as an iron source was shown to induce colitis and site-specific tumors in mice [30]. All these findings suggest *Lcn2* as a control element protecting against intestinal inflammation and tumorigenesis associated with alteration in the microbiota.

We now show that mice lacking LCN2 display a persistent colonization with the commensal bacteria SFB. Sequencing of PCR products revealed that the SFB in the ileum of LCN2 belong to the SFB species *Candidatus Arthromitus sp.* (not shown). SFB are a group of Gram-positive, host-adapted, spore forming organisms that attach to the absorptive intestinal epithelium (Figure 10) [38]. These bacteria are related to the genus *Clostridium*, which was elevated in both sexes of *Lcn2* null mice compared to wild type mice (Figure 6). They have the capacity to develop into long filaments, which are divided by transverse septa. Moreover, they have fundamental importance for the generation of innate and acquired immunity in the gastrointestinal and respiratory tracts. As such, these bacteria are essential in the maturation of the host gut immune barrier [39]. It has been further reported that the colonization with SFB in mice increases at around 20 d and is highest shortly after weaning and thereafter declines to the adult levels [39,40]. Therefore, it was unexpected to find such high amounts of SFB in the knockout animals compared to the lack or low amounts in age-matched controls (Figure 10 and Figure 11).

Several factors were identified that might regulate the transient nature of the SFB colonization, and one of these factors is the concentration of IgA. IgA is secreted across the intestinal epithelium where it binds to bacteria and limits direct contact with the host [41]. Colonization and adhesion of SFB to intestinal epithelial cells (IECs) lead to the induction and enhancement of the intestinal IgA responses [42,43]. In turn, IgA regulates the abundance of SFB so that a reduction in IgA causes an expansion of SFBs [44,45,46].

Moreover, histological images showed a deep penetration of SFB into the epithelial layer of cells (Figure 10). For the initiation of SFB replication and differentiation lifecycle leading to filamentation, epithelial tissue binding is essential (Figure 10) [47]. SFB attach to the host enterocytes in the mucous membrane of the epithelium without penetrating the host cell wall [48]. Nevertheless, this tight association with epithelial cells does itself not cause an inflammatory response in the *lamina propria* [49].

It is currently unknown if LCN2 deficiency directly impacts, for example, the receptors by which SFB contact the epithelial cells. It has to be noted here that SFB attach, in addition to absorptive epithelia, also to goblet cells and M-cells, which mutually do not express LCN2 [50]. On the other hand, LCN2 loss may change the antimicrobial response of epithelial cells, mucus constitution or set up of the antimicrobial barrier function, which confers a strong benefit for SFB colonization. Nevertheless, the observed changes in the other genera of the microbiome in *Lcn2*-deficient animals may directly impart a growth advantage for SFB, because they might be more competitive in the new environment compared to the other members of the new bacterial community.

Sequencing of SFB genomes suggested a particularly strong requirement for iron uptake since six different open reading frames (ORFs) for iron transporters were found in the mouse genome as well as three ORFs for ferric iron regulator family proteins [51]. Therefore, in the absence of LCN2, the better iron supply may favor SFB growth.

As outlined above, it has been reported that SFB have a critical impact on the development of the host immune system, including adaptive and innate components [52,53]. We show here that specific mMC proteases but not ctMC proteases are highly expressed in the distal ileum of *Lcn2*-deficient mice compared to control mice (Figure 9). We tried to confirm these findings by either immunohistochemistry using specific chymase antibodies or by Alcian blue and chloroacetate esterase staining. However, our efforts were not successful so far, most likely because the specimens in our laboratory are routinely fixed in formalin and mMC might be sensitive to this fixative [54]. However, MC, similar to dendritic cells or macrophages, may aid in shaping adaptive immune responses via their antigen-presenting ability.

In particular, the differentiation of naïve CD4^+^ T cells to form antigen-specific Th17 CD4^+^ cells is promoted by SFB [47]. We demonstrated that markers of Th17 cell polarization were higher expressed in *Lcn2* knockout mice compared to wild type mice. In particular, the transcription factor RORγt, as well as the cytokines IL-17A and IL-21 were significantly upregulated (Figure 11). Amongst other functions, IL-17A modulates neutrophil activation, differentiation, and chemotaxis as well as host defense peptide production [53]. Systemic depletion of neutrophils in mice caused increased production of IL-17A and ileal SFB colonization [55]. LCN2 is highly expressed in neutrophils and therefore, a quantitative or qualitative change in neutrophil function by loss of LCN2 might favor SFB growth as a secondary effect [56]. The signals to achieve Th17 differentiation are in part generated by the SFB-bound epithelial cells and involve intestinal macrophages and intestinal dendritic cells [57,58]. Binding of SFB to the host epithelial cells via the holdfast structure leads to a rearrangement of the actin cytoskeleton, induction of SAA, and production of reactive oxygen species (ROS), which aids in mediating Th17 cell differentiation [53,59]. In addition, binding of SFB induces expression of inducible NOS2 involved in ROS generation and elevated expression of RegIIIγ, representing a secreted, directly bactericidal C-type lectin that specifically targets Gram-positive bacteria [53,60]. The depletion of RegIIIγ, however, does not impact the total bacterial load of the small intestine but is involved in setting up a separation zone between Gram-positive bacteria and the epithelial cell layer of the villi [61]. In the absence of RegIIIγ, the mucosal bacterial load is much higher, implicating innate immune cells via the Toll-like receptor (TLR)/Myeloid differentiation factor 88 (MyD88)/RegIIIγ axis in the generation of a physical barrier between microbiota and host in the small intestine [61].

Accompanied by the gross changes in the microbiome, we found higher inflammatory activity in the small intestinal villi of *Lcn2*-deficient mice (Figure 9). This was paralleled by an increased expression of cytokines involved in inflammation such as IL-1β. Consistent with this finding, SFB flagellins have been shown to increase IL-1β expression in vivo in the small intestine, and IL-1β in turn increases Th17 cell differentiation [62,63]. Interestingly, the application of SFB flagellins to mice in vivo or to IECs in vitro strongly induces the expression of LCN2, showing that there is a direct connection of SFB and *Lcn2* expression [62]. This upregulation can be facilitated by the SFB-mediated increase in cytokines including TNF-α and IL-1β that can be a defense mechanism, similar to the induction of RegIIIγ counteracting pathogen invasion [64]. Therefore, in the absence of LCN2, one of the defense responses is missing, possibly favoring a prolonged colonization with SFB. As a fact underpinning several of the above described findings, it was shown that the deletion of the NF-κB transcription factor c-Rel causes reduced IgA generation, loss of T helper cell responses, impairment of IL-17A and IL-21 expression, and expansion of SFB [46].

Finally, this results in an alteration of the microbial composition seen as a kind of dysbiosis in the microbiomes of murine intestines.

It will now be of fundamental importance to clarify which LCN2-responsive receptors and pathways are involved in mediating this bypass, and further, to unravel the mechanisms by which LCN2 impacts the general concentration of individual bacteria. In addition, it will be interesting to clarify if the modulated colonization in *Lcn2*^-/-^ and wild type mice provokes differences in SFB-dependent IgA expression reported before [41,42,43,44,45,65,66,67,68]. Mice colonized with SFB were shown to be more resistant against fungal infections, underpinning the notion that SFB are important for the functionality of the adaptive immune response machinery [39].

In addition, the impact of iron and the LCN2/siderophore network on aspects of bacterial dysbiosis and SFB colonization needs further investigation. Iron is an essential element not only for higher organisms but also for most microorganisms. Bacterial iron acquisition through siderophores can be very efficient and their secretion often significantly contributes to virulence in infected hosts [69,70,71,72]. Due to their critical role for pathogens, siderophores serve as targets for the innate immune system to suppress bacterial growth. It is well accepted that one of the most important proteins in this context is LCN2 that is therefore also called siderocalin. The respective mechanism constitutes the so-called hypoferremia or anemia of infection that results in a significant decrease in the total iron concentration in body fluids following infection by a pathogen [69,70,71,72]. Consequently, this mechanism is interrupted in *Lcn2*-deficient mice and therefore enables the growth of siderophore-producing microbes such as *Escherichia*, *Staphylococcus*, *Prevotella*, and *Clostridium* [69,70,71,72].

The way in which LCN2 provokes changes in the expression of individual cytokines or chemokines (Figure 9C) and if higher expression of LCN2 is directly linked to the pathogenesis of inflammatory diseases of the gastrointestinal tract will be further evaluated in future studies. In particular, reports demonstrating that LCN2 is increased in the gut during the pathogenesis of CD [9] suggest that LCN2 can directly trigger the expression of inflammatory mediators.

## 4. Materials and Methods

### 4.1. Animals

*Lcn2*-deficient mice [26] and wild type (WT) mice on a C57BL/6J genetic background were kept in the same room with constant humidity (50%) and temperature (20 °C) under a 12:12 light-dark cycle. The animals had free access to tap water and food *ad libitum*. Depending on sex and genotype, the experimental animals were categorized into the following four experimental groups: male WT mice (n = 3), female WT mice (n = 5), male *Lcn2*^-/-^ mice (n = 2), and female *Lcn2*^-/-^ mice (n = 3) (Appendix A). The animals were fed with 10 mm pellets of a well-balanced nutrient (ssniff^®^ R/M-H special diet, #V1534, ssniff Spezialdiäten, Soest, Germany) containing a moderate energy density (gross energy: 16.3 MJ/kg, metabolizable energy 12.8 MJ/kg) and a very low nitrosamine content designed for rodents in maintenance metabolism and suitable for long-lasting experiments. The precise composition of this diet as provided by the supplier is given in Appendix A.

### 4.2. Collection of Fecal Samples

Sampling of fecal samples was carried out sterilely after the feces were removed from the animals. The feces were collected from the cage using clean tweezers (and gloves) and, if present, impurities such as litter or hair were removed. Samples from animals that were kept in the same cage (same sex and genotype) were pooled in a sterile centrifuge tube and kept at −80 °C until further processing. Since the availability of the fecal samples differed between the four groups, we processed the entire material and adjusted the number of sequencing samples accordingly (3–7 per group).

### 4.3. Cultivation of Microorganisms from Murine Feces Samples

For cultivation of microorganisms from murine feces samples, freshly extracted samples were transferred to 4 °C and 1.5 mL physiological saline solution per 0.35 g feces added. Thereafter, the samples were incubated for two hours at room temperature and homogenized by rigorous vortexing. Subsequently, 20–30 μL of the samples was streaked on different ready prepared agar plates with well-defined nutrient media to obtain pure colonies. As culture media, we used Columbia blood agar with sheep blood (PB5039A, Oxoid Limited, ThermoFisher Scientific, Schwerte, Germany), Müller Hinton agar with horse blood and 10 mg/L NAD (PB1229A, Oxoid Limited), MacConkey agar no. 3 (PO0495, Oxoid), and Sabouraud glucose agar with gentamicin and chloramphenicol (PA-254039.07, BD Biosciences, Heidelberg, Germany). Culture plates containing blood and Müller Hinton agar plates were incubated microaerobically (5% CO_2_, 37 °C), whereas MacConkey plates were incubated at 37 °C under aerobic CO_2_ conditions. Sabouraud agar plates were incubated for one week aerobically at room temperature. Forming colonies with a different appearance (color, shape, size, metabolic characteristics) were inoculated as pure cultures on fresh plates. In order to obtain single colonies, bacteria were spread with a sterile inoculation loop on the corresponding medium using the three-line technique.

### 4.4. Liquid Culture of Unknown Isolates

Liquid cultures were grown under both aerobic and microaerobic conditions. To establish pure cultures from colonies formed on solid media, individual clones were picked and inoculated with inoculation loops in liquid soybean casein digest broth (TSB) medium obtained from Merck Millipore (Darmstadt, Germany). Aerobic cultures were established in Erlenmeyer flasks that were filled to 20% of their nominal volume and cultured at 37 °C and 180 rpm in a shaking incubator until an adequate cell density was achieved. For microaerobic culture conditions, the Erlenmeyer flasks were filled to 50% of the nominal volume, and cultures were then incubated at 37 °C and 5% CO_2_.

### 4.5. Identification of Cultured Microorganisms Using MALDI-TOF MS

For identification of cultured microorganisms by MALDI-TOF-MS, a small amount of a single colony was picked with a sterile toothpick and embedded in 1 μL α-Cyano-4-hydroxycinnamic acid (CHCA) matrix and crystallized for 10 min. Each sample was applied in duplicate to increase the validity of the data. For identification of respective microorganisms, the embedded isolates were explosively vaporized in high vacuum by laser irradiation in a VITEK^®^ MS mass spectrometer (bioMérieux, Nürtingen, Germany). Resulting peptides were separated according to their mass-to-charge (m/z) ratio and determined in a time of flight (TOF) mass analyzer. Based on the TOF information, a characteristic total peptide mass fingerprint (PMF) was generated. The mass range of 2–20 kDa of the spectra were then used for microorganism identification using data from the microbial reference isolates stored in the VITEK^®^ MS system internal database.

### 4.6. Preparation of Small Intestinal Fecal Smears and Gram-Stains

The content of the distal ileum of the small intestine was gently extracted to not affect the epithelial integrity. The content was kept on ice and a part was resuspended in normal saline. One drop of this suspension was placed on a microscope slide and smeared with a second slide. The suspension was air dried, Methanol fixed for 5 min, air dried, and Gram-stained following routine procedures [73]. Preparations were mounted using DPX.

### 4.7. DNA Extraction from Murine Feces Samples and Quantification of the Total Bacterial Load

Prior to extraction, the feces within the individual groups were mixed properly, and 0.1 g of each mixture was then placed in a 2 mL Eppendorf tube and mixed with 1 mL 0.9% NaCl solution. Subsequently, the samples were incubated on ice for one hour, vortexed regularly, and then frozen at −20 °C overnight. After thawing, the samples were vortexed thoroughly for one minute and centrifuged at 2000 rpm to sediment unwanted faecal fibres and particles. Total DNA was then isolated from 200 μL of the supernatants using the PowerLyzer^®^ PowerSoil^®^ DNA isolation kit (Qiagen, Hilden, Germany) following the manufacturer’s instructions. The level of *EntA*-expressing gut bacteria was analyzed by qPCR using primers *EntA*-F and *EntA*-R [6].

### 4.8. Extraction of Chromosomal DNA from Bacterial Isolates

For extraction of chromosomal DNA from isolated bacteria, 2–4 mL of respective overnight cultures was centrifuged at 13,000 rpm for 5 min at room temperature. DNA was then isolated from the resulting pellet using the column-based PureLink^®^ Genomic DNA mini kit (Invitrogen, ThermoFisher Scientific) using the protocol specifications for Gram-positive cell lysates. The purified DNA was then eluted in 75 μL of DNase-free water and quantified by the Qubit assay (ThermoFisher Scientific), and the integrity of purified DNA was qualified by using Bioanalyzer DNA kits & reagents for the Bioanalyzer Automated Electrophoresis systems from Agilent (Agilent Technologies, Waldbronn, Germany). Finally, the purified DNA was stored at −20 °C until final use.

### 4.9. 16S rRNA Amplicon and Index PCR

For NGS, both the 16S rRNA amplicon (V4 region) PCR and the index PCR were performed according to the 16S Metagenomic Sequencing Library Preparation protocol of the Illumina MiSeq™ system (Illumina Inc., Munich, Germany). The cycling conditions of the GeneAmp^®^ PCR system 9700 (PE Applied Biosystems, ThermoFisher Scientific) were set as follows: initial denaturation: 95 °C for 3 min and 25 (16S rRNA amplicon) or 8 cycles (index) at 95 °C for 30 s, 55 °C for 30 s, and 72 °C for 30 s following a final elongation for 5 min at 72 °C. Final 16S rRNA sequencing was performed on the MiSeq™ platform using the PhiX library. The index primers used in our study are depicted in Appendix A. DNA was quantified as described above.

### 4.10. PCR Amplification of SFB 16S rDNA

For PCR amplification of 16S rDNA sequences, isolated bacterial DNA from the extracted small intestinal feces, 1 µL (10 pmol) of each primer, 1.25 U *Taq* DNA polymerase (Roche Diagnostics, Mannheim, Germany), and 0.5 µL dNTP (10 mmol each) were set up in a volume of 25 µL. The cycling conditions of the T3000 thermocycler (Biometra, Biometra GmbH, Göttingen, Germany) were set as follows: initial denaturation: 95 °C for 5 min and 40 cycles at 95 °C for 30 sec, 58 °C for 30 sec, and 72 °C for 45 s following a final elongation for 10 min at 72 °C. Fragments were separated in 1% agarose gels and extracted for sequencing. The primers used in this analysis are presented in Appendix A.

### 4.11. Bioinformatic and Statistical Data Analysis

The Illumina MiSeq™ system provides initial information during the 16S rRNA amplicon sequencing, which gives an indication of the quality and success of the run. In addition, a secondary on-instrument analysis of the sequencing data was done comparing the V4 amplicons with the Greengenes rRNA database [74]. In brief, the obtained sequences were grouped by barcode, and fastq files were generated. Then, paired ends were connected and the primer sequences were removed by the PANDAseq software [75]. The sequences were then filtered according to the expected amplicon length, ambiguous base assignment, and homopolymers with a length of more than eight nucleotides. The microbial analysis was then done with the open source software Mothur [76]. Multiple identical sequences were combined, and individual reads were compared with reference sequences of the SILVA rRNA database [77]. After removal of the chimeric and non-bacterial reads, the remaining sequences were taxonomically classified and summarized in so-called operational taxonomic units (OTUs). OTUs whose taxonomic identification was only possible to the family level were identified by genera with identical word stems using the NCBI Taxonomy Browser [78]. Contaminants that were identified in the negative controls were removed from the regular samples. QIIME [79] was used for the creation of PCoAs, and the 3D visualization was done with the EMPeror software [80,81]. Statistical evaluations were carried out with Microsoft Excel 2016. For statistics, we used the relative abundance of taxonomic groups as a normalization approach. β-diversity was calculated using the Jaccard index (*I_J_*) at the genus level. The mean relative abundances of each sample were summed groupwise to better illustrate group-related patterns and differences. The relative abundances of the individual samples at the genus level are depicted in Appendix A. For differential abundance analysis of promising genera, we used the discrete false-discovery rate (DS-FDR) [82]. We collapsed the classified OTUs at the genus level to run the test on log2-transformed data, testing for differences in mean values (*p* = 0.05, permutations = 10,000) with the standalone Python module [83].

### 4.12. RNA Expression Analysis

For RT-qPCR, total RNA from snap-frozen mouse distal small intestine tissue was extracted and purified with DNAse digestion as described before [84]. Synthesis of complementary DNA (cDNA) was done by reverse transcription of 1 μg of purified RNA in a final volume of 20 μL using Superscript II reverse transcriptase and random hexamer primers (all reagents from Invitrogen, Thermo Fisher Scientific). The synthesized cDNA was amplified in a 25 μL volume using SYBR GreenTM qPCR SuperMix (Applied Biosystems, Life Technologies, Darmstadt, Germany) as described [84]. All primers used for RT-qPCR were designed using the Universal Probe Library tool (Roche Diagnostics) and are listed in Appendix A. Relative levels of target mRNAs were quantified using the comparative CT method and the 2^−ΔΔCT^ method [85] and were normalized to the mRNA expression of GAPDH or β-actin. Final mRNA levels were expressed as the normalized quantity of the target transcript relative to the normalized quantity of the mRNA of the control group.

### 4.13. Immunohistochemistry and Tissue Staining

Tissue from the small intestine and colon was formalin-fixed and preserved through paraffin embedding. Sections of 5 µm thickness were stained by standard hematoxylin and eosin (H & E), periodic acid-Schiff (PAS) stain, chloracetate esterase stain (CAE), and Giemsa stain, following routine procedures [86,87,88]. Toluidine Blue stain of mouse tissue was performed as described in detail elsewhere [89]. For staining of CD45/leukocyte common antigen, antibody M0701 from Agilent (Agilent, Santa Clara, CA, USA) was used at a dilution of 1:2000. Cells of the monocyte lineages (e.g., macrophages, monocytes) were stained with an antibody directed against CD68/macrosialin (clone M0876, Agilent) used at a dilution of 1:400. Staining for the multiple myeloma oncogene 1 (MUM-1) was done for identification of plasma cells. The respective antibody M7259 was obtained from Agilent and applied at a dilution of 1:100. In addition, a peroxidase block (S2023, Agilent) and blocking of unspecific binding with mouse serum (Maus.SE.0010) obtained from Biosell (Bio & Sell, Feucht/Nürnberg, Germany) were done. The final detection was done using the Envison/Flex kit (Agilent).

### 4.14. Sodium Dodecylsulfate Polyacrylamide Gel Electrophoresis and Western Blot Analysis of Proteins from the Distal Small Intestine

A small piece of the distal intestine was directly placed in 200 µL radioimmunoprecipitation (RIPA) buffer (50 mM Tris-HCl (pH 7.4), 150 mM NaCl, 1% (*w*/*v*) NP-40, 0.1% (*w*/*v*) SDS, and 0.5% (*w*/*v*) sodium deoxycholate) containing the complete protease inhibitors (Roche Diagnostics) and phosphatase inhibitors (Sigma–Aldrich, Taufkirchen, Germany). The tissue was homogenized using a MM400 ball mill (Retsch, Haan, Germany) with a pre-cooled Eppendorf tube holder for 5 min at 30 Hz. Equal amounts of total cellular proteins (10–50 µg/lane) determined by the DC protein assay (Bio-Rad, Düsseldorf, Germany) were mixed with NuPAGE™ LDS electrophoresis sample buffer (Invitrogen, Thermo Fisher Scientific, Dreieich, Germany) and supplemented with 50 mM dithiothreitol (DTT) as a reducing reagent. Denaturation of protein samples, electroblotting, and Western blot analysis were essentially applied as described previously [90]. After blotting, proper transfer of proteins was demonstrated in Ponceau S stain, and unspecific binding sites were blocked with 5% (*w*/*v*) non-fat milk powder in Tris-buffered saline with Tween 20 (TBST). For detection of individual proteins, the primary antibodies for LCN2/NGAL (AF3508, R & D Systems, Abingdon, UK), GAPDH (6C5) (sc-32233, Santa Cruz, Biotech, Santa Cruz, CA, USA), β-actin (A5441, Sigma–Aldrich, MI, USA), and E-cadherin (AP20195PU-N, OriGene, Rockville, MD, USA) were diluted in 2.5% (*w*/*v*) non-fat milk powder in TBST. Primary antibodies were visualized with anti-mouse, anti-rabbit or anti-goat IgG secondary antibodies (all from Santa Cruz) with the SuperSignal chemiluminescent substrate (Pierce, Bonn, Germany).

## 5. Conclusions

In conclusion, our study underpins the notion that LCN2 is an important factor in innate immunity that prevents infections of bacterial pathogens by sequestering iron. When LCN2 is missing, lowered resistance to infection results and the survival, virulence, and propagation of opportunistic, potentially life-threatening microbiota exploiting the siderophore-mediated iron uptake are favored.

## Figures and Tables

**Figure 1 ijms-22-13156-f001:**
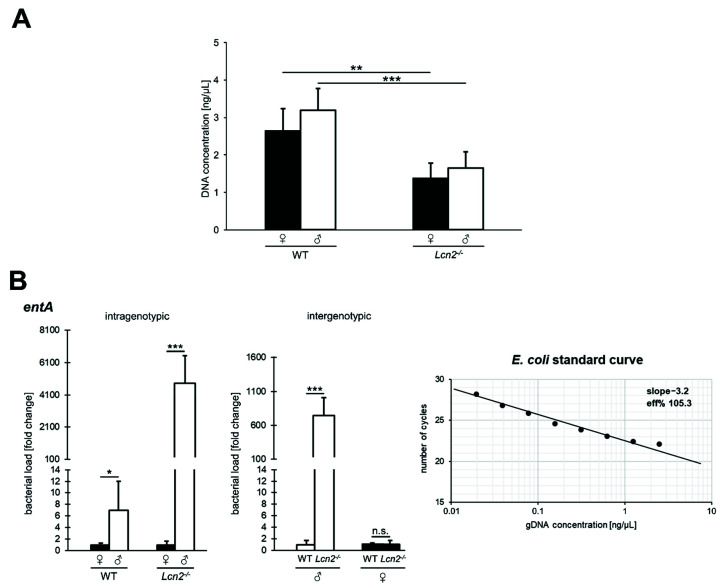
Concentration of total and intestinal DNA abundance in murine feces. (**A**) After extraction, the dsDNA concentration was fluorometrically determined by Qubit^®^ assay. The bars represent the arithmetic mean of the individual measurements (± SD; *Lcn2*^-/-^ (n = 7 or 5) and wild type mice (n = 7 or 3)). Statistical evaluation was performed by an unpaired *t*-test, whereby the difference of two samples from a level of *p* < 0.05 is considered statistically significant (**: *p* < 0.01, ***: *p* < 0.001). (**B**) Investigation of the intestinal abundance of *entA*-coding bacteria using qPCR. The data were normalized using an *E. coli* standard curve (*right*). Grouped bars represent the fold change ratio between two data sets (± SD; *Lcn2*^-/-^ (n = 7 or 5) and wild type mice (n = 7 or 3)), with the wild type defined as 100%. Statistical evaluation was performed by unpaired *t*-test, whereby the difference between two samples from a level of *p* < 0.05 is considered statistically significant (*: *p* < 0.05, ***: *p* < 0.001; n.s. = not significant; Eff% = efficiency [%]).

**Figure 2 ijms-22-13156-f002:**
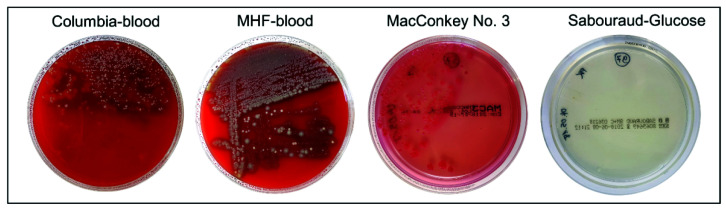
Cultivation of microorganisms from murine feces. Representative pictures of the heterogeneously overgrown Columbia and MHF blood agar (microaerobic, 37 °C), MacConkey No. 3 agar (aerobic, 37 °C), as well as the non-overgrown Sabouraud glucose agar (aerobic, room temperature) used to isolate individual bacterial species. From these and similar plates, colonies of different phenotypes were cultured in pure culture, isolated, and then identified by matrix-assisted laser desorption/ionization-time of flight (MALDI-TOF) mass spectrometry.

**Figure 3 ijms-22-13156-f003:**
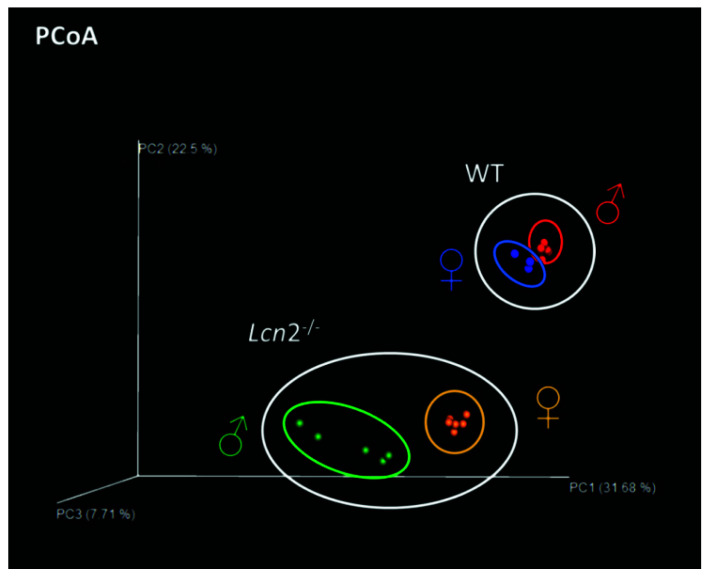
Principal coordinates analysis (PCoA) of the intestinal tract microbial community. Depicted are the unweighted UniFrac distance metrics between microbial communities of the intestinal tract of *Lcn2*^-/-^ and WT mice of both sexes. The three-dimensional visualization of the sequencing data was performed with the EMPeror software. Each point represents a single sample that is displayed in the same color within a group.

**Figure 4 ijms-22-13156-f004:**
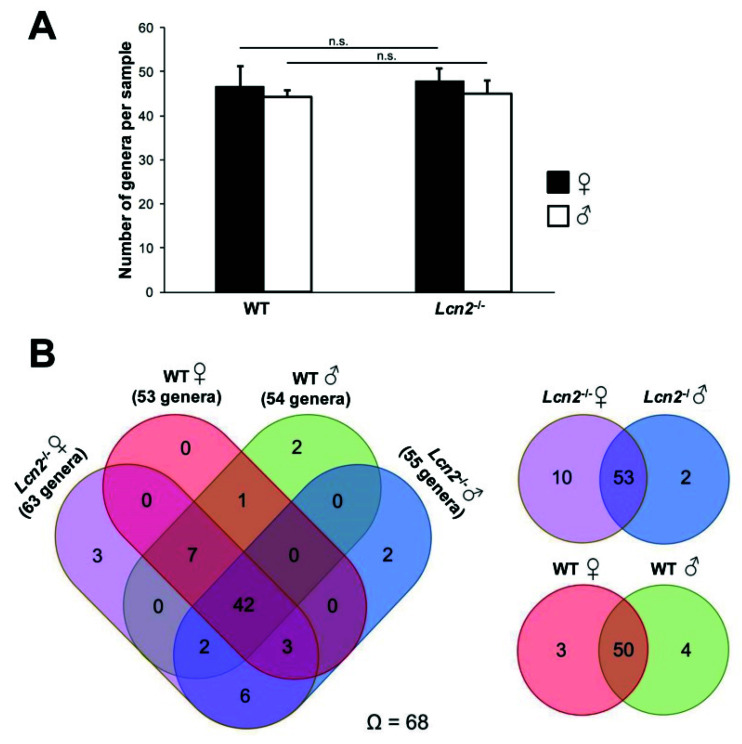
Gender-specific diversity and genus spectrum of the intestinal microbiome of *Lcn2*^-/-^ and wild type mice. (**A**) The bars represent the arithmetic mean of the genera per sample (± SD; *Lcn2*^-/-^ (n = 7 or 5) and wild type mice (n = 7 or 3)). Statistical evaluation was performed using the Kruskal–Wallis test, whereby the difference of two samples from a level of *p* < 0.05 is considered statistically significant (n.s. = not significant). (**B**) Venn diagram showing shared genera between the different groups by absolute numbers (Ω = total number of genera).

**Figure 5 ijms-22-13156-f005:**
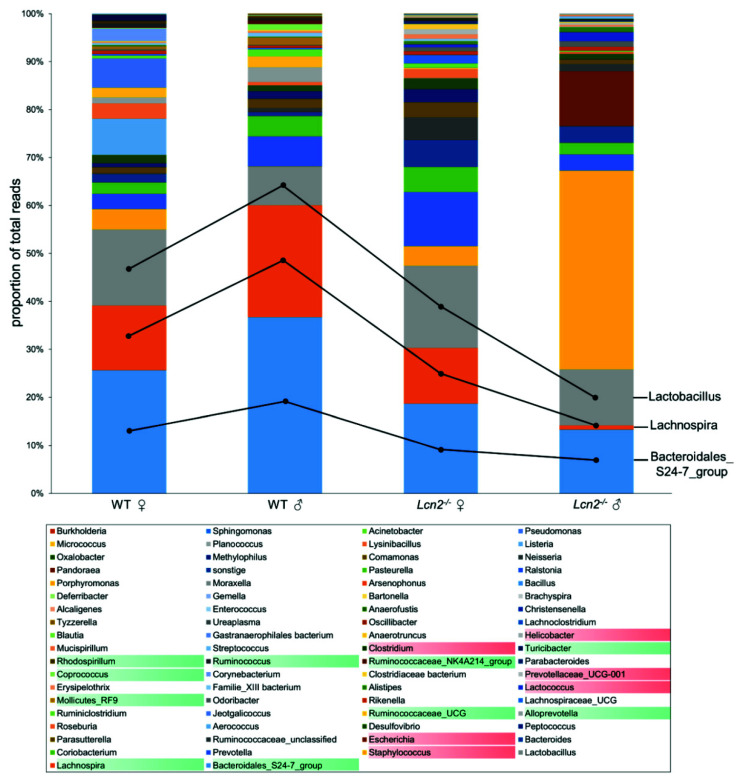
Relative abundance and distribution of identified genera of the intestinal microbiome of *Lcn2*^-/-^ and wild type mice. The mean of the relative abundance of genera is plotted groupwise as a percentage of total reads [%]. Genus depicted on a *green* or *red* background showed significant differences in WT and *Lcn2*^-/-^ mice (*green*: decrease or missing in *Lcn2*^-/-^; *red*: missing in WT or increased in *Lcn2*^-/-^). The reading direction begins with the *Bacteroidales_S24-7_group* and is shown as an example using the first three taxonomic units in the bar chart.

**Figure 6 ijms-22-13156-f006:**
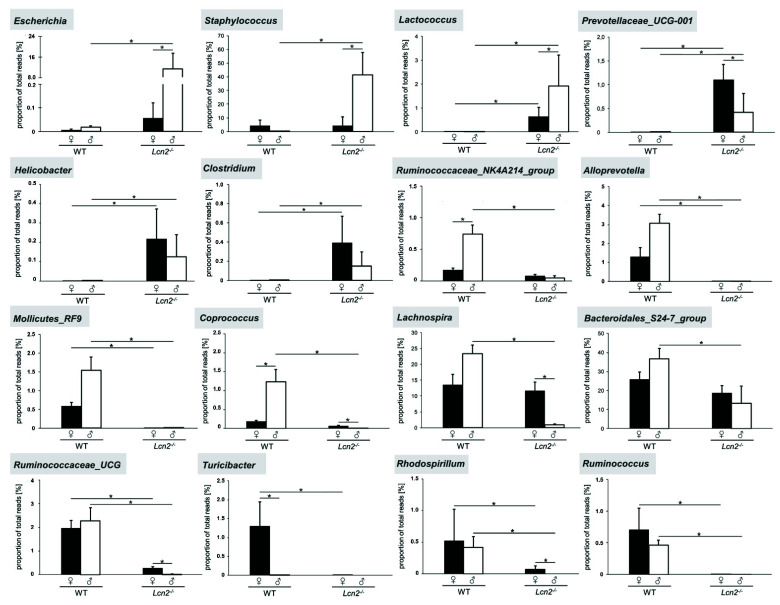
Relative abundance of conspicuously altered genera between WT and *Lcn2*^-/-^ mice. The relative abundance of the genera is plotted as a percentage of the total reads [%]. The bars in each diagram represent the arithmetic mean of biological multiple approaches (±SD; *Lcn2*^-/-^ (n = 7 or 5) and wild type mice (n = 7 or 3)). Depicted are differences in abundance of *Escherichia*, *Staphylococcus*, *Lactococcus*, *Prevotellaceae_UCG-001, Helicobacter*, *Clostridium*, the *Ruminococcaceae_NK4A214_group*, *Alloprevotella*, *Mollicutes_RF9*, *Coprococcus*, *Lachnospira*, the *Bacteroidales_S24-7_group*, *Ruminococcaceae_UCG*, *Turicibacter*, *Rhodospirillum*, and *Ruminococcus*. Statistical evaluation was performed using the discrete false-discovery rate (DS-FDR), whereby the difference between two samples from a level of *p* < 0.05 is considered statistically significant (*: *p* < 0.05).

**Figure 7 ijms-22-13156-f007:**
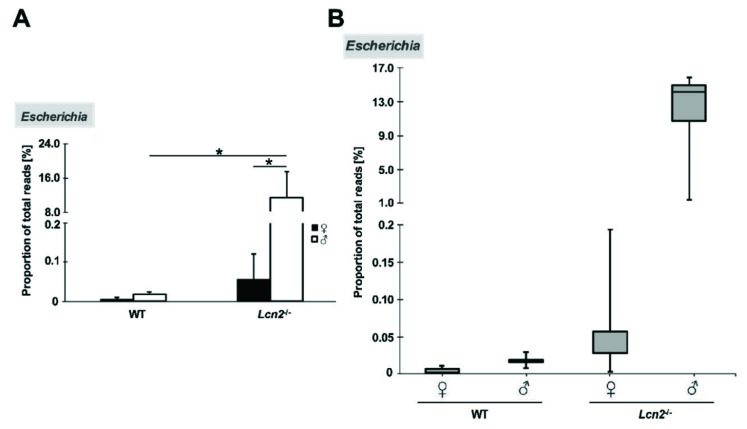
Frequency distribution of the data using the example of the genus *Escherichia*. (**A**) Percentage of total reads [%] in the different genders and genotypes (*Lcn2*^-/-^ (n = 7 or 5) and wild type mice (n = 7 or 3)). (**B**) The relative abundance is plotted as a percentage of the total reads [%]; 50% of the mean data lie within the boxes. In the box plot, the whiskers range from the quartiles to the minimum or maximum including outliers is shown. *: *p* < 0.05.

**Figure 8 ijms-22-13156-f008:**
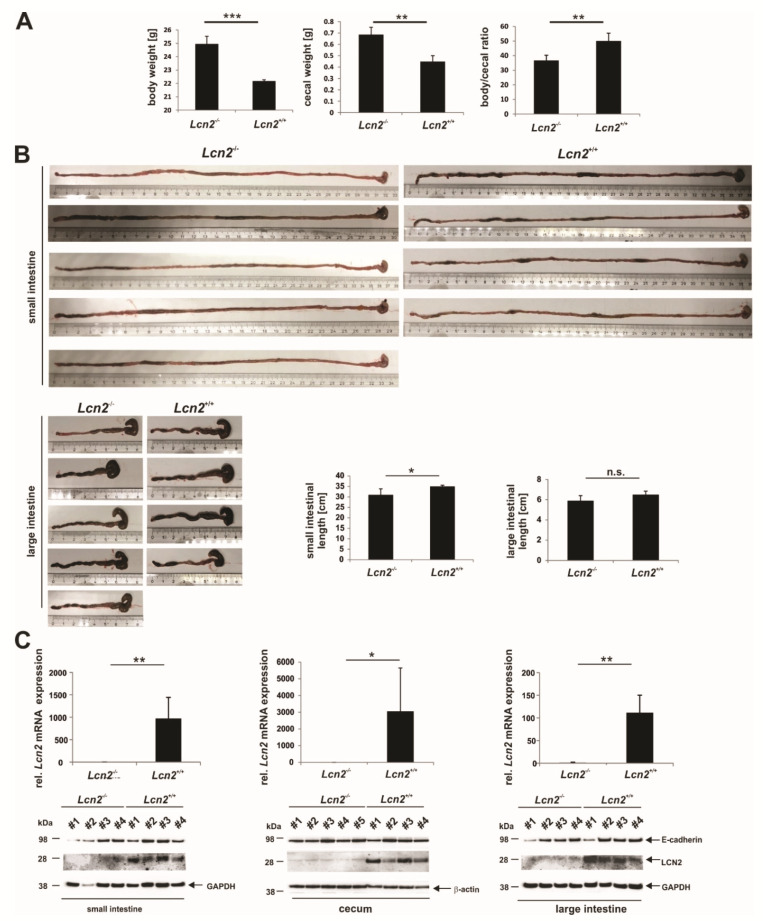
Comparative analysis of the intestine macroscopical parameters and expression of *Lcn2* in wild type (n = 4) and *Lcn2*^-/-^ mice (n = 4 or 5). (**A**) Evaluation of the body weight of male animals at the same age (left). Corresponding ceca were prepared and weighed (middle). Ratio of body to cecal weight (right). (**B**) Gross macroscopical and graphical comparison of the total small and large intestinal lengths measured (n = 4 or 5). (**C**) Quantitative analysis of *Lcn2* mRNA (upper) and LCN2 protein expression (lower) in the distal small intestine (left) cecum (middle), and distal colon (right) of wild type (*Lcn2*^+/+^) and *Lcn2* null mice (each n = 4 or 5). Relative mRNA expression of respective genes was done by RT-qPCR and normalized to *Gapdh or β-actin*. Equal protein loading was demonstrated by re-probing the membranes with a specific antibody directed against GAPDH or β-actin. E-cadherin was probed as a marker for epithelial-to-mesenchymal transition (EMT). Primers used are given in Appendix A. Statistical evaluation in (**A**–**C**) was performed by an unpaired *t*-test, whereby the difference of two samples from a level of *p* < 0.05 is considered statistically significant (*: *p* < 0.05, **: *p* < 0.01, ***: *p* < 0.001).

**Figure 9 ijms-22-13156-f009:**
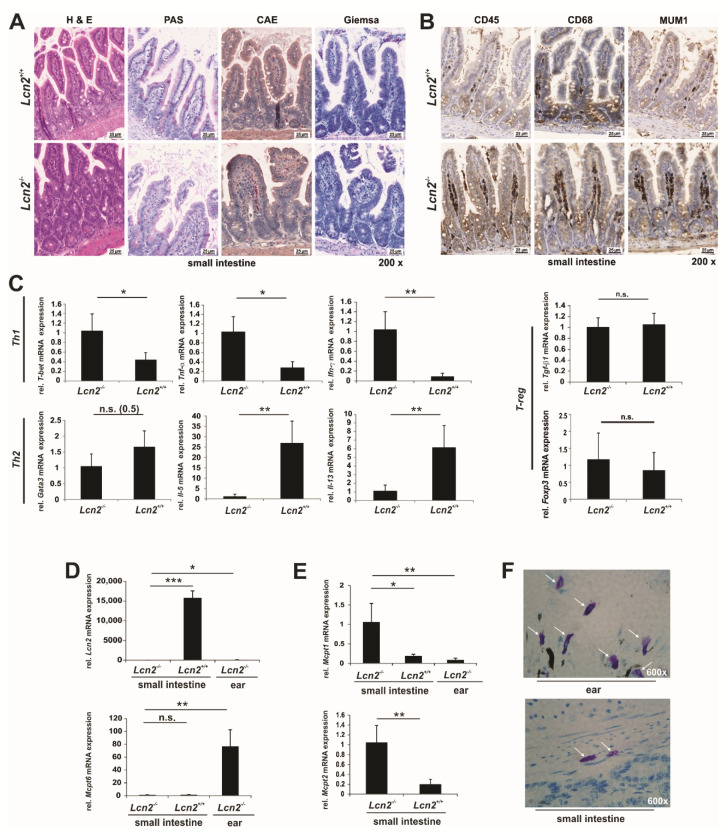
Comparative analysis of inflammatory-related parameters in the small intestine. (**A**) Paraffin-embedded tissue samples prepared from the small intestine of wild type and *Lcn2*^-/-^ mice were stained with hematoxylin & eosin (H & E), Periodic acid-Schiff (PAS), chloroacetate esterase (CAE), and Giemsa (each n = 4). (**B**) Comparative analysis of inflammatory activity in the ileum of *Lcn2* null and wild type mice. Individual sections from the small intestine were stained with antibodies directed against CD45, MUM-1, and CD68. (**C**) Quantitative mRNA expression analysis of representative markers for Th1 cells (*T-bet Tnf-α*, *Ifn-γ*), Th2 cells (*Gata3*, *Il-5*, *Il-13*), and T_reg_ cells (*Tgf-β1*, *Foxp3*) in the distal small intestine of wild type and *Lcn2* null mice (each n = 4). (**D**,**E**) Quantitative mRNA expression analysis of *Lcn2, Mcpt6* (marker of connective tissue MC), and *Mcpt1/Mcpt2* (markers of mucosal MC) in the distal small intestine of wild type and *Lcn2*^-/-^ mice (n = 4). The relative mRNA expression of respective genes was done by RT-qPCR and normalized to β-*actin*. Primers used are given in Appendix A. Statistical evaluation in (**C**–**E**) was performed by an unpaired *t*-test, whereby the difference of two samples from a level of *p* < 0.05 is considered statistically significant (*: *p* < 0.05, **: *p* < 0.01, ***: *p* < 0.001). (**F**) Representative Toluidine Blue stain of paraffin-embedded tissue samples prepared from the ear or small intestine of *Lcn2* null mice. White arrows indicate connective tissue mast cells.

**Figure 10 ijms-22-13156-f010:**
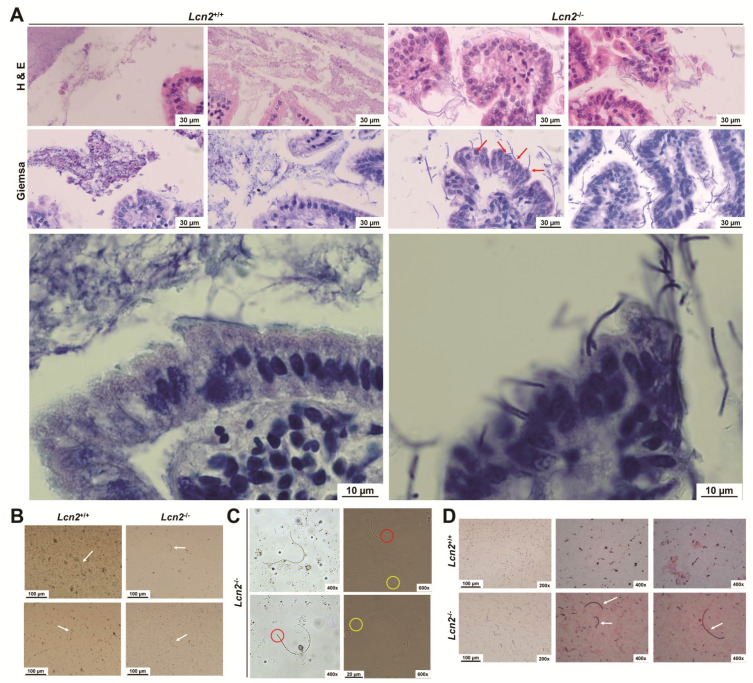
Detection of segmented filamentous bacteria in the distal small intestine of wild type and *Lcn2*^-/-^ mice. (**A**) Paraffin-embedded tissue samples prepared from the small intestine of wild type and *Lcn2*^-/-^ mice were stained with hematoxylin & eosin (H & E), and Giemsa (each n = 4). Red arrows indicate the close relationship of segmented filamentous bacteria (SFB) and epithelial cells. Images taken at higher magnification (630×) are shown in the lower panels. (**B**,**C**) Smears of extracted small intestinal contents from wild type and *Lcn2*^-/-^ mice were mounted in DPX and used for microscopical analysis. White arrows indicate SFB (**B**). Red circles indicate the bold and yellow circles the fine segmented tip of SFB (**C**). (**D**) Smears of extracted small intestinal contents from wild type and *Lcn2*^-/-^ mice were prepared and stained according to the Gram procedure, mounted in DPX, and used for microscopy. White arrows indicate Gram-stained SFB.

**Figure 11 ijms-22-13156-f011:**
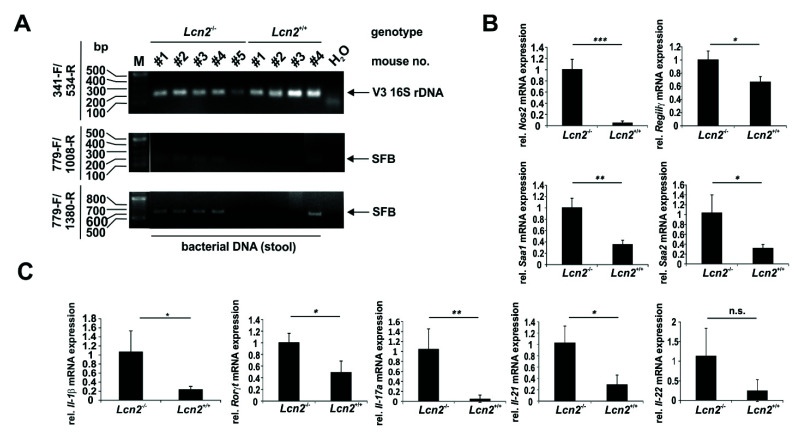
Proof and consequences of elevated quantities of segmented filamentous bacteria (SFB) in *Lcn2*^-/-^ mice. (**A**) DNA was extracted from the distal ileal content of wild type and *Lcn2*^-/-^ mice and analyzed by PCR for bacterial V3 16S- and SFB-specific 16S rRNAs. (**B**,**C**) Quantitative mRNA expression analysis of *Nos2*, *RegIIIγ*, *Saa1*, *Saa2, Il-1β, Rorγt, Il-17a*, *Il-21*, and *Il-22* in the distal small intestine of wild type (*Lcn2*^+/+^) and *Lcn2* null mice (each n = 4). Relative mRNA expression of respective genes was done by RT-qPCR and normalized to *β-actin*. Primers used are given in Appendix A. *: *p* < 0.05, **: *p* < 0.01, ***: *p* < 0.001.

**Figure 12 ijms-22-13156-f012:**
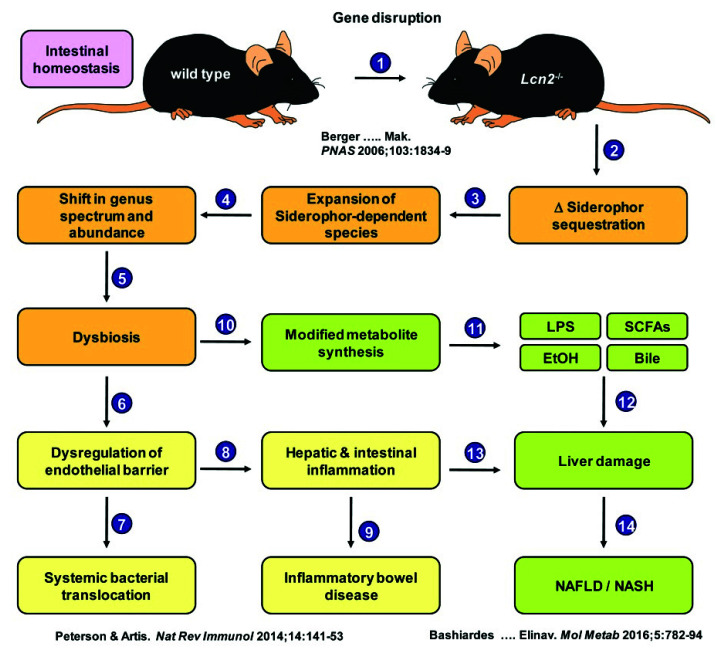
Model hypothesis of the effect of *Lcn2* loss on the intestinal microbiome and the resulting systemic consequences. Abbreviations used are EtOH, ethanol; LPS, lipopolysaccharides; SCFAs, short-chain fatty acids. Relevant references supporting this model hypothesis are depicted in the graph and are [26,36,37].

**Table 1 ijms-22-13156-t001:** Identification of aerobically- and microaerobically-cultured bacterial species from murine fecal samples using MALDI-TOF mass spectrometry and 16S rDNA sequence analysis.

Species	Wild Type (♀)	Wild Type (♂)	*Lcn2*^-/-^ (♀)	*Lcn2*^-/-^ (♂)
*Escherichia coli*	M (99.9)	M (99.9)	M (99.9)	M (99.9)
*Staphylococcus xylosus*	M (99.9)	M (99.9)	M (99.9)	M (99.9)
*Aerococcus viridans*	M (99.9)	M (99.9)	M (99.9)	M (99.9)
*Enterococcus faecalis*		M (99.9)		
*Lactobacillus murinus/* *Lactobacillus animalis*		S (99.0)		
*Lactobacillus johnsonii*			S (98.0)	
*Lactococcus formosensis*				S (99.0)
*Streptococcus danieliae*				S (99.0)

Note: In the groups marked with “S” (sequencing) and “M” (MALDI-TOF), the corresponding species could be identified. The confidence value [%] and the sequence identity [%] are shown in parentheses and provide information on the validity of the results. Values equal to or larger than 70.0 indicate a successful identification.

## Data Availability

The data presented in this study and additional data underpinning depicted findings are available on request from the corresponding authors.

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
