# Peer review of "Depletion of Lipocalin 2 (LCN2) in Mice Leads to Dysbiosis and Persistent Colonization with Segmented Filamentous Bacteria"

_ijms, 2021, doi:10.3390/ijms222313156_

Round 1
Reviewer 1 Report
The manuscript “Depletion of Lipocalin 2 (LCN2) in Mice Leads to Dysbiosis and Persistent Colonization with Segmented Filamentous Bacteria” by Patrick Klüber and coworkers addresses the very interesting topic on how lipocalin 2 (LCN2) depletion in mice alters their intestinal microbiome, leading to increased intestinal inflammatory activity and to a higher abundance of segmented filamentous bacteria (SFB) in the ileal epithelium. These findings have important implications for dysbiosis and for increased inflammatory activity in the intestine, which may be tied to several diseases, including Crohn’s disease. To this end, they combine multiple methods and analyses, ranging from cultivation of microorganisms from murine feces to identification of cultured microorganisms by MALDI-TOF MS, next-generation sequencing and analysis, and more.
Overall, this is a very nice and thorough study. The experiments are carefully designed, the results are well-described the discussion is good.
I only have a few minor comments:
1) The manuscript is generally well written. I have noted two typos that should be corrected. One is in the title, where “Persistant” should be “Persistent”. The second is in the title on line 202, where “homogen” seems to be truncated (homogenous?).
2) In several figures where quantification and statistical analysis are presented, the number of independent experiments is missing and should be mentioned. This includes Fig. 1A, B, Fig. 4A, Fig. 6, and Fig. 7. Moreover, in all these figures, it would be beneficial if the actual data points representing each individual experiment would be depicted on top of the bar plots.
3) In Figures 8 and 9, the number of independent experiments is given, but only the average values and the SD values are shown. It would be good to add the actual individual data points where relevant.
Author Response
Dear reviewer,
please see our comments to your review in the pdf file attached.

Reviewer 2 Report
The manuscript by Klüber and co-authors describes the effects of Lipocalin 2 deficiency in the composition of the gut microbiota as well as in the host immune response. The bacterial composition and changes are characterized by qPCR, culture dependent and independent methods. The assessment of the immune response is performed by tissue staining and mRNA expression of immune cell markers and cytokines. Although the multiple approaches used to investigate the changes at both the microbiome and the immune system are noteworthy, the sequential presentation of the results is not always clear as if multiple stories were put together in the manuscript. I have also uncertainties over some aspects of the methodology that should be addressed to strengthen the conclusions drawn by the authors.
In particular:
- a great emphasis is given in the first part of the manuscript to the gender-dependent differences in the bacterial composition, but the gender issue appears neglected in the evaluation of immune response and colonization of SFB. It would be expected that gender-dependent differences in microbial composition result in gender-dependent changes in the immune response in the gut. Why did the authors choose not to investigate gender-dependent differences in the immune response?
- it is sometimes unclear how the results from the different approaches fit together, for instance it is unclear to me how the results from qPCR for entA expression, culture-dependent methods and NGS could be combined together to explain the effects of LCN2 deficiency on the microbiome. I would say that the increased abundance of intestinal bacteria encoding entA in male knock-out mice is driven by Escherischia, isolated from the four groups (Table 1) and more expressed in male knock-out mice (Figure 6). What is the significance of the changes that the authors observe in the composition of the other genera, mainly belonging to the phyla Firmicutes and Bacteroidetes, for instance Staphylococcus or Clostridium? I think that the authors should provide an interpretation of the main differentially expressed genera in terms of dysbiosis and their hypothesis on the systemic consequences.
- the methodology also appears unclear in some instances. I found difficult to reconcile the description of fecal collection in 4.2 (samples from animals … were pooled …), DNA extraction in 4.7 (feces from individual groups were mixed) and the number of animals in 4.1 and Table S2 (n = 2-5) with the results presented, for instance, in Figure 3 or Figure S1. Could the authors please clarify the number of samples that were sequenced and analyzed?
- the methodology of microbiome analysis should be better detailed. For instance, in Table S4, the authors list negative controls, but it is not explained how the authors dealt with contaminants from the negative controls. Line 221: “Statistical evaluation was performed by unpaired t-test, [..]”: t-test is a parametric test which assumes alpha diversity distribution at genus level to be Normal. This assumption has not been verified. Moreover, the number of samples used for the t-test of alpha index is not adequate since some groups have only three samples and all groups have less than 6 samples. This reviewer suggests to use another diversity index which has the same information content on the number of common/not common genera: the beta-diversity Jaccard index at genus level. A Kruskall Wallis test between groups solve the parametric test problem and the Kruskall Wallis test between groups on diversity evaluated within each group could compensate for the lack of samples. Moreover, a rarefaction approach is usually applied when comparing diversity indexes. Please specify if any rarefaction has been applied. If rarefaction has not been applied, comments on effects of sequencing depth of each sample and corresponding diversity indexes should be added. Lines 223-224: “Similarities between the different groups are shown in the corresponding intersections of the Venn diagram. The genus spectrum is indicated by absolute numbers (Ω = total number of genera).”: this sentence should be revised. In the metagenomic analysis context, the number of genera in a group of samples is usually the alpha diversity index evaluated at genus level. The Venn diagram here can indicate only the number of genera common to two groups. If this is a similarity index, please define it in the paper. Lines 237-238: “The summary of these sequencing results are depicted graphically in Figure 5.”; please, specify what kind of summary is reported in Figure 5. In particular, it seems, but not reported in any part of the paper, that authors have grouped together sequencing of each group. The grouping criterion should be explicitly indicated (common criteria are sum, mean or median of relative or absolute abundances) and justified to the scope of the analysis. Finally, it is unclear why an unpaired t test was used to test the individual differences in Figure 6. Since data generated by NGS sequencing are intrinsically compositional, proportions between samples or groups should not be compared directly with standard statistical methods, but other methodologies (i.e. Aldex2, ANCOM, …..) should be used (see, for instance, 10.5808/GI.2019.17.1.e6, 10.1093/bioinformatics/bty175, 10.1038/s41467-019-10656-5).
- Lines 237-242 are unclear to me.
- In figure 9, the authors state that the inflammatory activity is increased in Lcn2 null mice by tissue staining, and attempt to corroborate this statement with RT-qPCR of cytokines. The results are not straightforward, as Il6 and Il10 levels are reduced and increased, respectively, in knock-out mice. The authors should provide protein quantification of key cytokines to confirm the higher inflammatory activity.
- line 475: “favoring fungal growth”; in the acknowledgements “panfungal PCR” is mentioned. Where are these results shown?
Author Response
Dear reviewer,
please find our comments to your review in the attached pdf file.

Round 2
Reviewer 2 Report
The authors satisfactorily addressed most of my comments.
However, I still have concerns over some aspects of the metagenomics analysis, primarily on the use of the t test between groups of proportion of selected genera. The application of the t test to metagenomic data is questionable since sequencing outcome is not quantitative. It is preferable to use Aldex2 (that, as correctly stated by the authors, applies to all genera) and then verify if Aldex gives a significant difference for the genera of interest and the corresponding effect-size, or otherwise test significant differences between groups by using ratio between proportions (ratio between genera of interest and a genus common to all samples). Also, if rarefaction has not been used as normalization method (sequencing depth of each samples should be added in the paper in order to emphasize that diversity is not affected by differences in sampling depth between samples), sum of relative abundances, as reported by the authors in Figure 5, cannot be applied.
Author Response
Dear Reviewer 2,
please see our response to your valid comments in the attachment.
